# Scalable Structure Learning of Continuous-Time Bayesian Networks from Incomplete Data

**Dominik Linzner**[1]   **Michael Schmidt**[1]   **Heinz Koeppl**[1,2]
[1]Department of Electrical Engineering and Information Technology
[2]Department of Biology
Technische Universität Darmstadt
{dominik.linzner, michael.schmidt, heinz.koeppl}@bcs.tu-darmstadt.de

## Abstract

Continuous-time Bayesian Networks (CTBNs) represent a compact yet powerful framework for understanding multivariate time-series data. Given complete data, parameters and structure can be estimated efficiently in closed-form. However, if data is incomplete, the latent states of the CTBN have to be estimated by laboriously simulating the intractable dynamics of the assumed CTBN. This is a problem, especially for structure learning tasks, where this has to be done for each element of a super-exponentially growing set of possible structures. In order to circumvent this notorious bottleneck, we develop a novel gradient-based approach to structure learning. Instead of sampling and scoring all possible structures individually, we assume the generator of the CTBN to be composed as a mixture of generators stemming from different structures. In this framework, structure learning can be performed via a gradient-based optimization of mixture weights. We combine this approach with a new variational method that allows for a closed-form calculation of this mixture marginal likelihood. We show the scalability of our method by learning structures of previously inaccessible sizes from synthetic and real-world data.

## 1   Introduction

Learning correlative or causative dependencies in multivariate data is a fundamental problem in science and has application across many disciplines such as natural and social sciences, finance and engineering [1, 20]. Most statistical approaches consider the case of snapshot or static data, where one assumes that the data is drawn from an unknown probability distribution. For that case several methods for learning the directed or undirected dependency structure have been proposed, e.g., the PC algorithm [21, 13] or the graphical LASSO [8, 12], respectively. Causality for such models can only partially be recovered up to an equivalence class that relates to the preservation of v-structures [21] in the graphical model corresponding to the distribution. If longitudinal and especially temporal data is available, structure learning methods need to exploit the temporal ordering of cause and effect that is implicit in the data for determining the causal dependency structure. One assumes that the data are drawn from an unknown stochastic process. Classical approaches such as Granger causality or transfer entropy methods usually require large sample sizes [23]. Dynamic Bayesian networks offer an appealing framework to formulate structure learning for temporal data within the graphical model framework [10]. The fact that the time granularity of the data can often be very different from the actual granularity of the underlying process motivates the extension to continuous-time Bayesian networks (CTBN) [14], where no time granularity of the unknown process has to be assumed. Learning the structure within the CTBN framework involves a combinatorial search over structures and is hence generally limited to low-dimensional problems even if one considers variational approaches [11] and/or greedy hill-climbing strategies in structure space [15, 16]. Reminiscent of

optimization-based approaches such as graphical LASSO, where structure scoring is circumvented by performing gradient descent on the edge coefficients of the structure under a sparsity constraint, we here propose the first gradient-based scheme for learning the structure of CTBNs.

## 2 Background

### 2.1 Continuous-time Bayesian Networks

We consider continuous-time Markov chains (CTMCs) $\{X(t)\}_{t\geq 0}$ taking values in a countable state-space $\mathcal{S}$. A time-homogeneous Markov chain evolves according to an intensity matrix $R : \mathcal{S} \times \mathcal{S} \rightarrow \mathbb{R}$, whose elements are denoted by $R(s, s')$, where $s, s' \in \mathcal{S}$. A continuous-time Bayesian network [14] is defined as an $N$-component process over a factorized state-space $\mathcal{S} = \mathcal{X}_1 \times \cdots \times \mathcal{X}_N$ evolving jointly as a CTMC. For local states $x_i, x_i' \in \mathcal{X}_i$, we will drop the states' component index $i$, if evident by the context and no ambiguity arises. We impose a directed graph structure $\mathcal{G} = (V, E)$, encoding the relationship among the components $V \equiv \{V_1, \ldots, V_N\}$, which we refer to as nodes. These are connected via an edge set $E \subseteq V \times V$. This quantity is the structure, which we will later learn. The state of each component is denoted by $X_i(t)$ assuming values in $\mathcal{X}_i$, which depends only on the states of a subset of nodes, called the parent set $\mathrm{par}_{\mathcal{G}}(i) \equiv \{j \mid (j, i) \in E\}$. Conversely, we define the child set $\mathrm{ch}_{\mathcal{G}}(i) \equiv \{j \mid (i, j) \in E\}$. The dynamics of a local state $X_i(t)$ are described as a Markov process conditioned on the current state of all its parents $U_n(t)$ taking values in $\mathcal{U}_i \equiv \{\mathcal{X}_j \mid j \in \mathrm{par}_{\mathcal{G}}(i)\}$. They can then be expressed by means of the conditional intensity matrices (CIMs) $R_i : \mathcal{X}_i \times \mathcal{X}_i \times \mathcal{U}_i \rightarrow \mathbb{R}$, where $u_i \equiv (u_1, \ldots u_L) \in \mathcal{U}_i$ denotes the current state of the parents ($L = |\mathrm{par}_{\mathcal{G}}(i)|$). The CIMs are the generators of the dynamics of a CTBN. Specifically, we can express the probability of finding node $i$ in state $x'$ after some small time-step $h$, given that it was in state $x$ at time $t$ with $x, x' \in \mathcal{X}_i$ as

$$p(X_i(t + h) = x' \mid X_i(t) = x, U_i(t) = u) = \delta_{x,x'} + hR_i(x, x' \mid u) + o(h),$$

where $R_i(x, x' \mid u)$ is the rate the transition $x \rightarrow x'$ given the parents' state $u \in \mathcal{U}_i$ and $\delta_{x,x'}$ being the Kronecker-delta. We further make use of the small $o(h)$ notation, which is defined via $\lim_{h\to 0} o(h)/h = 0$. It holds that $R_i(x, x \mid u) = -\sum_{x' \neq x} R_i(x, x' \mid u)$. The CIMs are connected to the joint intensity matrix $R$ of the CTMC via amalgamation – see, for example [14].

### 2.2 Structure Learning for CTBNs

**Complete data.** The likelihood of a CTBN can be expressed in terms of its sufficient statistics [15], $M_i(x, x' \mid u)$, which denotes the number of transitions of node $i$ from state $x$ to $x'$ and $T_i(x \mid u)$, which denotes the amount of time node $i$ spend in state $x$. In order to avoid clutter, we introduce the sets $\mathcal{M} \equiv \{M_i(x, x' \mid u) \mid i \in \{1, \ldots, N\}, x, x' \in \mathcal{X}, u \in \mathcal{U}\}$ and $\mathcal{T} \equiv \{T_i(x \mid u) \mid i \in \{1, \ldots, N\}, x \in \mathcal{X}, u \in \mathcal{U}\}$. The likelihood then takes the form

$$p(\mathcal{M}, \mathcal{T} \mid \mathcal{G}, R) = \prod_{i=1}^{N} \exp\left\{ \sum_{x,x' \neq x,u} M_i(x, x' \mid u) \ln R_i(x, x' \mid u) - T_i(x \mid u)R_i(x, x' \mid u) \right\}. \tag{1}$$

In [15] and similarly in [22] it was shown that a marginal likelihood for the structure can be calculated in closed form, when assuming a gamma prior over the rates $R_i(x, x' \mid u) \sim \mathrm{Gam}(\alpha_i(x, x' \mid u), \beta_i(x' \mid u))$. In this case, the marginal log-likelihood of a structure takes the form

$$\ln p(\mathcal{M}, \mathcal{T} \mid \mathcal{G}, \alpha, \beta) \propto \sum_{i=1}^{N} \sum_{u,x,x' \neq x} \left\{ \ln \Gamma\left(\bar{\alpha}_i(x, x' \mid u)\right) - \bar{\alpha}_i(x, x' \mid u) \ln \bar{\beta}_i(x \mid u) \right\}, \tag{2}$$

with $\bar{\alpha}_i(x, x' \mid u) \equiv M_i(x, x' \mid u) + \alpha_i(x, x' \mid u)$ and $\bar{\beta}_i(x \mid u) \equiv T_i(x \mid u) + \beta_i(x \mid u)$. Structure learning in previous works [16, 22, 11] is then performed by iterating over possible structures and scoring them using the marginal likelihood. The best scoring structure is then the *maximum-a-posteriori* estimate of the structure.

**Incomplete data.** In many cases, the sufficient statistics of a CTBN cannot be provided. Instead, data comes in the form of noisy state observations at some points in time. In the following, we

will assume data is provided in form of $N_s$ samples $\mathcal{D} \equiv \{(t^k, y^k) \mid k \in \{1, \ldots, N_s\}\}$, where $y^k$ is some, possibly noisy, measurement of the latent-state generated by some observation model $y^k \sim p(Y = y^k \mid X(t^k) = s)$ at time $t^k$. This data is incomplete, as the sufficient statistics of the underlying latent process have to be estimated before model identification can be performed. In [16], an expectation-maximization for structure learning (SEM) was introduced, in which, given a proposal CTBN, sufficient statistics were first estimated by exact inference, the CTBN parameters were optimized given those expected sufficient-statistics and, subsequently, structures where scored via (1). Similarly, in [11] expected sufficient-statistics were estimated via variational inference under marginal (parameter-free) dynamics and structures were then scored via (2).

The problem of structure learning from incomplete data has two distinct bottlenecks, $(i)$ Latent state estimation (scales exponentially in the number of nodes) $(ii)$ Structure identification (scales super-exponentially in the number of nodes). While bottleneck $(i)$ has been tackled in many ways [4, 5, 19, 11], existing approaches [16, 11] employ a combinatorial search over structures, thus an efficient solution for bottleneck $(ii)$ is still outstanding.

**Our approach.** We will employ a similar strategy as mentioned above in this manuscript. However, statistics are estimated under a marginal CTBN that no longer depends on rate parameters or a discrete structure. Instead, statistics are estimated given a mixture of different parent-sets. Thus, instead of blindly iterating over possible structures in a hill-climbing procedure, we can update our distribution over structures by a gradient step. This allows us to directly converge into regions of high-probability. Further, in combination of this gradient-based approach with a high-order variational method, we can perform estimation of the expected sufficient-statistics in large systems. These two features combined, enable us to perform structure learning in large systems. An implementation of our method is available via Git[1].

## 3 Likelihood of CTBNs Under a Mixture of CIMs

**Complete data.** In the following, we consider a CTBN over some [2]over-complete graph $\mathcal{G}$. In practice, this graph may be derived from data as prior knowledge. In the absence of prior knowledge, we will choose the full graph. We want to represent its CIMs $R_i(x, x' \mid u)$, here for node $i$, as mixture of CIMs of smaller support and write by using the power-set $\mathcal{P}(\cdot)$ (set of all possible subsets)

$$R_i(x, x' \mid u) = \sum_{m \in \mathcal{P}(\mathrm{par}_{\mathcal{G}}(i))} \pi_i(m) r_i(x, x' \mid u_m) \equiv \mathsf{E}_i^\pi [r_i(x, x' \mid u_m)], \tag{3}$$

where $u_m$ denotes the projection of the full parent-state $u$ on the subset $m$, i.e. $f(u_m) = \sum_{u/u_m} f(u)$, and the expectation $\mathsf{E}_i^\pi [f(\theta_m)] = \sum_{m \in \mathcal{P}(\mathrm{par}_{\mathcal{G}}(i))} \pi_i(m) f(\theta_m)$. The mixture-weights are given by a distribution $\pi_i \in \Delta_i$ with $\Delta_i$ being the $|\mathcal{P}(\mathrm{par}_{\mathcal{G}}(i))|-$dimensional probability simplex. Corresponding edge probabilities of the graph can be computed via marginalization. The probability that an edge $e_{ij} \in E$ exists is then

$$p(e_{ij} = 1) = \sum_{m \in \mathcal{P}(\mathrm{par}_{\mathcal{G}}(j))} \pi_j(m) \mathbb{1}(i \in m), \tag{4}$$

with $\mathbb{1}(\cdot)$ being the indicator function. In order to arrive at a marginal score for the mixture we insert (3) into (1) and apply Jensen's inequality $\mathsf{E}_i^\pi [\ln(r)] \leq \ln(\mathsf{E}_i^\pi [r])$. This yields a lower-bound to the mixture likelihood

$$p(\mathcal{M}, \mathcal{T} \mid \pi, r) \geq \prod_{i=1}^{N} \prod_{x, x' \neq x, u_m} e^{\mathsf{E}_i^\pi \left[ M_i(x, x' | u_m) \ln r_i(x, x' | u_m) - T_i(x | u_m) r_i(x, x' | u_m) \right]}.$$

For details on this derivation, we refer to the supplementary material A.1. Note that Jensens inequality, which only provides a poor approximation in general, improves with increasing concentration of probability mass and becomes exact for degenerate distributions. For the task of selecting a CTBN with a specific parent-set, it is useful to marginalize over the rate parameters $r$ of the CTBNs. This allows for a direct estimation of the parent-set, without first estimating the rates. This

marginal likelihood can be computed under the assumption of independent gamma prior distributions $r_i(x, x' \mid u_m) \sim \mathrm{Gam}(\alpha_i(x, x' \mid u_m), \beta_i(x' \mid u_m))$ over the rates. The marginal likelihood lower-bound can then be computed analytically. Under the assumption of independent Dirichlet priors $\pi_i \sim \mathrm{Dir}(\pi_i \mid c_i)$, with concentration parameters $c_i$ we arrive at a lower-bound to the marginal log-posterior of the mixture weights $\pi$

$$\ln p(\pi \mid \mathcal{M}, \mathcal{T}, \alpha, \beta) \geq \sum_i \mathcal{F}_i[\mathcal{M}, \mathcal{T}, \pi] + \ln Z, \tag{5}$$

$$\mathcal{F}_i[\mathcal{M}, \mathcal{T}, \pi] \equiv \sum_{m, u_m, x, x' \neq x} \left\{ \ln \Gamma \left( \bar{\alpha}_i(x, x' \mid u_m) \right) - \bar{\alpha}_i(x, x' \mid u_m) \ln \bar{\beta}_i(x \mid u_m) \right\} + \ln \mathrm{Dir}(\pi_i \mid c_i),$$

with the updated posterior parameters $\bar{\alpha}_i(x, x' \mid u_m) \equiv \pi_i(m) M_i(x, x' \mid u_m) + \alpha_i(x, x' \mid u_m)$ and $\bar{\beta}_i(x \mid u_m) \equiv \pi_i(m) T_i(x \mid u_m) + \beta_i(x \mid u_m)$. For details, we refer to the supplementary material A.2. The constant log-partition function $\ln Z$ can be ignored in the following analysis. Because (5) decomposes into a sum of node-wise terms, the *maximum-a-posterior* estimate of the mixture weights of node $i$ can be calculated as solution of the following optimization problem:

$$\pi_i^* = \underset{\pi_i \in \Delta_i}{\arg\max} \left\{ \mathcal{F}_i[\mathcal{M}, \mathcal{T}, \pi] \right\}. \tag{6}$$

By construction, learning the mixture weights $\pi$ of the CIMs, corresponds to learning a distribution over parent-sets for each node. We thus re-expressed the problem of structure learning to an estimation of $\pi$. Further, we note that for any degenerate $\pi$, (5) coincides with the exact structure score (2).

**Incomplete data.** In the case of incomplete noisy data $\mathcal{D}$, the likelihood of the CTBN does no longer decompose into node-wise terms. Instead, the likelihood is one of the full amalgamated CTMC [16]. In order to tackle this problem, approximation methods through sampling [7, 6, 19], or variational approaches [4, 5] have been investigated. These, however, either fail to treat high-dimensional spaces because of sample sparsity, are unsatisfactory in terms of accuracy, or provide only an uncontrolled approximation. Our method is based on a variational approximation, e.g. weak coupling expansion [11]. Under this approximation, we recover by the same calculation an approximate likelihood of the same form as (1), where the sufficient statistics $M_i(x, x' \mid u)$ and $T_i(x \mid u)$ are, however, replaced by their expectation $\mathsf{E}_q\left[M_i(x, x' \mid u)\right]$ and $\mathsf{E}_q\left[T_i(x \mid u)\right]$ under a variational distribution $q$, – for details we refer to the supplementary B.1. Subsequently, also our optimization objective $\mathcal{F}_i[\mathcal{M}, \mathcal{T}, \pi]$ becomes dependent on the variational distribution $\mathcal{F}_i[\mathcal{D}, \pi, q]$. In the following chapter, we will develop an Expectation-Maximization (EM)-algorithm that iteratively estimates the expected sufficient-statistics given the mixture-weights and subsequently optimizes those mixture-weights given the expected sufficient-statistics.

## 4 Incomplete data: Expected Sufficient Statistics Under a Mixture of CIMs

**Short review of the foundational method.** In [11], the exact posterior over paths of a CTBN given incomplete data $\mathcal{D}$, is approximated by a path measure $q(X_{[0,T]})$ of a variational time-inhomogeneous Markov process via a higher order variational inference method. For a CTBN, this path measure is fully described by its node-wise marginals $q_i(x', x, u; t) \equiv q_i(X_i(t+h) = x', X_i(t) = x, U_i(t) = u; t)$. From it, one can compute the marginal probability $q_i(x; t)$ of node $i$ to be in state $x$, the marginal probability of the parents $q_i(U_i(t) = u; t) \equiv q_i^u(t)$ and the marginal transition probability $\tau_i(x, x', u; t) \equiv \lim_{h \to 0} q_i(x', x, u; t)/h$ for $x \neq x'$. The exact form of the expected statistics were calculated to be

$$\mathsf{E}_q\left[T_i(x \mid u)\right] \equiv \int_0^T \mathrm{d}t \, q_i(x; t) q_i^u(t), \quad \mathsf{E}_q\left[M_i(x, x' \mid u)\right] \equiv \int_0^T \mathrm{d}t \, \tau_i(x, x', u; t). \tag{7}$$

In the following, we will use the short-hand $\mathsf{E}_q\left[\mathcal{M}\right]$ and $\mathsf{E}_q\left[\mathcal{T}\right]$ to denote the sets of expected sufficient-statistics. We note, that the variational distribution $q$ has the support of the full over-complete parent-set $\mathrm{par}_{\mathcal{G}}(i)$. Via marginalization of $q_i(x', x, u; t)$, the marginal probability and the marginal transition probability can be shown to be connected via the relation

$$\frac{\mathrm{d}}{\mathrm{d}t} q_i(x; t) = \sum_{x' \neq x, u} \left[ \tau_i(x,' x, u; t) - \tau_i(x, x', u; t) \right]. \tag{8}$$

---

**Algorithm 1** Stationary points of Euler–Lagrange equation

---

1: **Input:** Initial trajectories $q_i(x;t)$, boundary conditions $q(x;0)$ and $\rho(x;T)$, mixture weights $\pi$ and data $\mathcal{D}$.
2: **repeat**
3:    **repeat**
4:       **for all** $i \in \{1, \ldots, N\}$ **do**
5:          **for all** $(y^k, t^k) \in \mathcal{D}$ **do**
6:             Update $\rho_i(t)$ by backward propagation from $t_k$ to $t_{k-1}$ using (10) fulfilling the jump conditons (12).
7:          **end for**
8:          Update $q_i(t)$ by forward propagation using (10) given $\rho_i(t)$.
9:       **end for**
10:    **until** Convergence
11:    Compute expected sufficient statistics using (7) and (11) from $q_i(t)$ and $\rho_i(t)$.
12: **until** Convergence of $\mathcal{F}[\mathcal{D}, \pi, q]$
13: **Output:** Set of expected sufficient statistics $\mathsf{E}_q[\mathcal{M}]$ and $\mathsf{E}_q[\mathcal{T}]$.

---

**Application to our setting.** As discussed in the last section, the objective function in the incomplete data case has the same form as (5)

$$\mathcal{F}_i[\mathcal{D}, \pi, q] \equiv \sum_{m, u_m, x, x' \neq x} \left\{ \ln \Gamma \left( \bar{\alpha}_i^q(x, x' \mid u_m) \right) - \bar{\alpha}_i^q(x, x' \mid u_m) \ln \bar{\beta}_i^q(x \mid u_m) \right\} + \ln \mathrm{Dir}(\pi_i \mid c_i),$$

(9)

however, now with $\bar{\alpha}_i^q(x, x' \mid u_m) \equiv \pi_i(m) \mathsf{E}_q[M_i(x, x' \mid u_m)] + \alpha_i(x, x' \mid u_m)$ and $\bar{\beta}_i^q(x \mid u_m) \equiv \pi_i(m) \mathsf{E}_q[T_i(x \mid u_m)] + \beta_i(x \mid u_m)$. In order to arrive at approximation to the expected sufficient statistics in our case, we have to maximize (9) with respect to $q$, while fulfilling the constraint (8). The corresponding Lagrangian becomes

$$\mathcal{L}[\mathcal{D}, \pi, q, \lambda] =$$
$$\sum_{i=1}^N \left[ \mathcal{F}_i[\mathcal{D}, \pi, q] - \sum_{x, x' \neq x, u} \int_0^T \mathrm{d}t \, \lambda_i(x;t) \left\{ \frac{\mathrm{d}}{\mathrm{d}t} q_i(x;t) - [\tau_i(x,' x, u; t) - \tau_i(x, x', u; t)] \right\} \right],$$

with Lagrange-multipliers $\lambda_i(x;t)$. In order to derive Euler-Lagrange equations, we employ *Stirlings-approximation* for the gamma function $\Gamma(z) = \sqrt{\frac{2\pi}{z}} \left( \frac{z}{e} \right)^z + \mathcal{O}\left( \frac{1}{z} \right)$, which becomes exact asymptotically. In our case, Stirlings-approximation is valid if $\bar{\alpha} \gg 1$. We thereby assumed that either enough data has been recorded, or a sufficiently strong prior $\alpha$. Finally, we recover the approximate forward- and backward-equations of the mixture CTBNs as the stationary point of the Lagrangian *Euler-Lagrange equations*

$$\frac{\mathrm{d}}{\mathrm{d}t} \rho_i(t) = \tilde{\Omega}_i^\pi(t) \rho_i(t), \quad \frac{\mathrm{d}}{\mathrm{d}t} q_i(t) = q_i(t) \Omega_i^\pi(t),$$

(10)

with effective rate matrices

$$\Omega_i^\pi(x, x'; t) \equiv \mathsf{E}_i^u \left[ \tilde{R}_i^\pi(x, x' \mid u) \right] \frac{\rho_i(x'; t)}{\rho_i(x; t)}$$

$$\tilde{\Omega}_i^\pi(x, x'; t) \equiv (1 - \delta_{x, x'}) \mathsf{E}_i^u \left[ \tilde{R}_i^\pi(x, x' \mid u) \right] + \delta_{x, x'} \left\{ \mathsf{E}_i^u \left[ R_i^\pi(x, x' \mid u) \right] + \Psi_i(x; t) \right\},$$

with $\rho_i(x; t) \equiv \exp(-\lambda_i(x; t))$ and $\Psi_i(x; t)$ as given in the supplementary material B.2. Further we have introduced the shorthand $\mathsf{E}_i^u[f(u)] = \sum_u f(u) q_i^u(t)$

and defined the posterior expected rates

$$R_i^\pi(x, x' \mid u) \equiv \mathsf{E}_i^\pi \left[ \frac{\bar{\alpha}_i^q(x, x' \mid u_m)}{\bar{\beta}_i^q(x \mid u_m)} \right], \quad \tilde{R}_i^\pi(x, x' \mid u) \equiv \prod_m \left( \frac{\bar{\alpha}_i^q(x, x' \mid u_m)}{\bar{\beta}_i^q(x \mid u_m)} \right)^{\pi_i(m)},$$

---

**Algorithm 2** Gradient-based Structure Learning

---
1: **Input:** Initial trajectories $q_i(x;t)$, boundary conditions $q_i(x;0)$ and $\rho_i(x;T)$, initial mixture weights $\pi^{(0)}$, data $\mathcal{D}$ and iterator $n = 0$
2: **repeat**
3:     Compute expected sufficient statistics $\mathsf{E}_q[\mathcal{M}]$ and $\mathsf{E}_q[\mathcal{T}]$ given $\pi^{(n)}$ using Algorithm 1.
4:     **for all** $i \in \{1, \ldots, N\}$ **do**
5:         Maximize (6) with respect to $\pi_i$, set maximizer $\pi_i^{(n+1)} = \pi_i^*$ and $n \to n+1$.
6:     **end for**
7: **until** Convergence of $\mathcal{F}[\mathcal{D}, \pi, q]$
8: **Output:** Maximum-a-posteriori mixture weights $\pi^{(n)}$

---

which take the form of an arithmetic and geometric mean, respectively. For the variational transition-matrix we find the algebraic relationship

$$\tau_i(x, x', u; t) = q_i(x;t)q_i^u(t)\tilde{R}_i^\pi(x, x' \mid u)\frac{\rho_i(x';t)}{\rho_i(x;t)}. \tag{11}$$

Because, the derivation is quite lengthy, we refer to supplementary B.2 for details. In order to incorporate noisy observations into the CTBN dynamics, we need to specify an observation model. In the following we assume that the data likelihood factorizes $p(Y = y^k \mid X(t_k) = s) = \prod_i p_i(Y_i = y_i^k \mid X_i(t_k) = x)$, allowing us to condition on the data by enforcing jump conditions

$$\lim_{t \to t^{k-}} \rho_i(x;t) = \lim_{t \to t^{k+}} p_i(Y_i = y_i^k \mid X_i(t_k) = x)\rho_i(x;t). \tag{12}$$

The converged solutions of the ODE system can then be used to compute the sufficient statistics via (7). For a full derivation, we refer to the supplementary material B.2.

We note that in the limiting case of a degenerate mixture distribution $\pi$, this set of equations reduces to the marginal dynamics for CTBNs proposed in [11]. The set of ODEs can be solved iteratively as a fixed-point procedure in the same manner as in previous works [17, 4] (see Algorithm 1) in a forward-backward procedure.

**Exhaustive structure search.** As we are now able to calculate expected-sufficient statistics given mixture weights $\pi$, we can design an EM-algorithm for structure learning. For this iteratively optimize $\pi$ given the expected sufficient statistics, which we subsequently re-calculate. The EM-algorithm is summarized in Algorithm 2. In contrast to the exact EM-procedure [16], we preserve structure modularity. We can thus optimize the parent-set of each node independently. This already provides a huge boost in performance, as in our case the search space scales exponentially in the components, instead of super-exponentially. In the paragraph "Greedy structure search", we will demonstrate how to further reduce complexity to a polynomial scaling, while preserving most prediction accuracy.

**Restricted exhaustive search.** In many cases, especially for applications in molecular biology, comprehensive [3]databases of putative interactions are available and can be used to construct over-complete yet not fully connected prior networks $\mathcal{G}_0$ of reported gene and protein interactions. In this case we can restrict the search space by excluding possible non-reported parents for every node $i$, $\mathrm{par}_\mathcal{G}(i) = \mathrm{par}_{\mathcal{G}_0}(i)$, allowing for structure learning of large networks.

**Greedy structure search.** Although we have derived a gradient-based scheme for exhaustive search, the number of possible mixture components still equals the number of all possible parent-sets. However, in many applications, it is reasonable to assume the number of parents to be limited, which corresponds to a sparsity assumption. For this reason, greedy schemes for structure learning have been proposed in previous works [16]. Here, candidate parent-sets were limited to have at most $K$ parents, in which case, the number of candidate graphs only grows polynomially in the number of nodes. In order to incorporate a similar scheme in our method, we have to perform an additional approximation to the set of equations (10). The problem lies in the expectation step (Algorithm 1), as expectation is performed with respect to the full over-complete graph. In order to calculate expectations of the geometric mean $\mathsf{E}_i^u[\tilde{R}_i^\pi(x, x' \mid u)]$, we have to consider the over-complete set of parenting nodes $q_i^u(t)$ for each node $i$. However, for the calculation of the arithmetic mean $\mathsf{E}_i^u[R_i^\pi(x, x' \mid u)]$ only

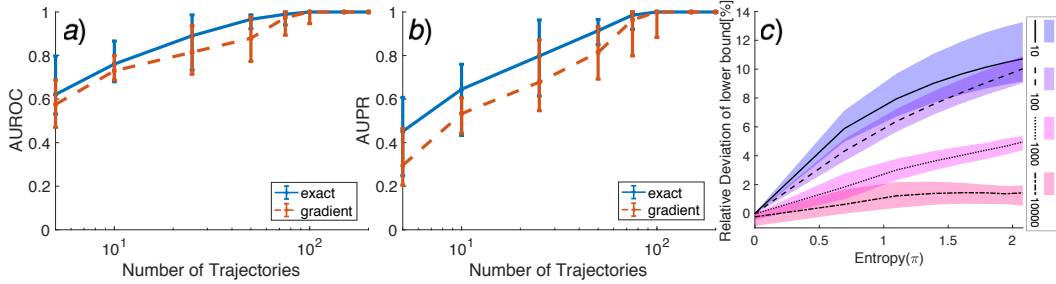

Figure 1: a) and b) AUROC and AUPR, respectively, for complete observations for different numbers of trajectories. Learning is performed via the graph-score (2) (blue) and gradient-based optimization of the marginal mixture likelihood (5) (red-dashed). c) Relative deviation of approximate marginal mixture likelihood (5) from the exact marginal likelihood, computed via numerical integration, for mixtures of different entropies given different amounts of trajectories (legend). Confidence intervals are given by 75% and 25% percentiles.

parent-sets restricted to the considered sub-graphs have to be considered, due to linearity. For this reason, we approximate the geometric mean by the arithmetic mean $\tilde{R}_i^\pi \approx R_i^\pi$, corresponding to the first-order expansion $\mathsf{E}_i^\pi[\ln(x)] = \ln(\mathsf{E}_i^\pi[x]) + \mathcal{O}(\mathsf{Var}[x])$, which, as before, becomes more valid for more concentrated $\pi_i$ and is exact if $\pi_i$ is degenerate.

## 5 Experiments

We demonstrate the effectiveness of our method on synthetic and two real-world data sets. For all experiments, we consider a fixed set of hyper-parameters. We set the Dirichlet concentration parameter $c_i = 0.9$ for all $i \in \{1, \ldots, N\}$. Further, we assume a prior for the generators, which is uninformative on the structure $\alpha_i(x, x' \mid u) = 5$ and $\beta_i(x \mid u) = 10$, for all $x, x' \in \mathcal{X}_i, u \in \mathcal{U}_i$. For the optimization step in Algorithm 2, we use standard Matlab implementation of the interior-point method with 100 random restarts. This is feasible, as the Jacobian of (9) can be calculated analytically.

### 5.1 Synthetic Data

In this experiment, we consider synthetic data generated by random graphs with a flat degree distribution, truncated at degree two, i.e. each nodes has a maximal number of two parents. We restrict the state-space of each node to be binary $\mathcal{X} = \{-1, 1\}$. The generators of each node are chosen such that they undergo Glauber-dynamics [9] $R_i(x, \bar{x} \mid u) = \frac{1}{2} + \frac{1}{2}\tanh\left(\gamma x \sum_{j \in \mathrm{par}_{\mathcal{G}}(i)} u_j\right)$, which is a popular model for benchmarking, also in CTBN literature [4]. The parameter $\gamma$ denotes the coupling-strength of node $j$ to $i$. With increasing $\gamma$ the dynamics of the network become increasingly deterministic, converging to a logical-model for $\gamma \to \infty$. In order to avoid violating the weak-coupling assumption [11], underlying our method, we choose $\gamma = 0.6$. We generated a varying number of trajectories with each containing 10 transitions. In order to have a fair evaluation, we generate data from thirty random graphs among five nodes, as described above. By computing the edge probabilities $p(e_{ij} = 1)$ via (4), we can evaluate the performance of our method as an edge-classifier by computing the receiver-operator characteristic curve (ROC) and the precision-recall curve (PR) and their area-under-curve (AUROC) and (AUPR). For an unbiased classifier, both quantities have to approach 1, for increasing amounts of data.

**Complete data.** In this experiment, we investigate the viability of using the marginal mixture likelihood lower-bound as in (5) given the complete data in the form of the sufficient statistics $\mathcal{M}$ and $\mathcal{T}$. In Figure 1 we compare the AUROCs a) and AUPRs b) achieved in an edge classification task using exhaustive scoring of the exact marginal likelihood (2) as in [15] (blue) and gradient ascend in $\pi$ of the mixture marginal likelihood lower-bound (red-dashed) as in (5). In Figure 1 c) we show via numerical integration, that the marginal mixture likelihood lower-bound approaches the exact one (2) for decreasing entropy of $\pi$ and increasing number of trajectories. Small negative deviations are due

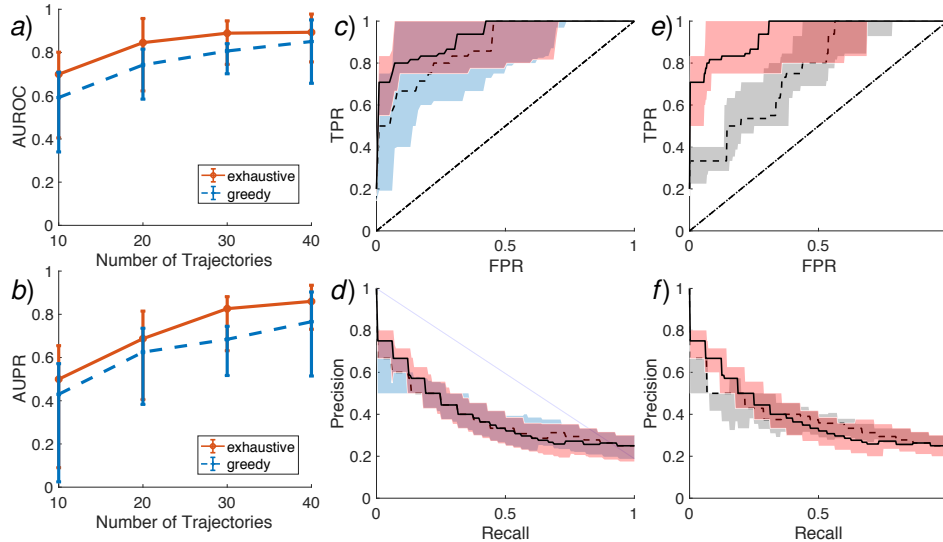

Figure 2: a) AUROCs and b) AUPRs for varying number of trajectories. c) ROC and d) PR curve for 40 trajectories. In all plots (red) denotes the exhaustive, (blue/dashed) the greedy-algorithm. e) ROC-curve f) PR-curve for different initial $\pi^{(0)}$, where (red) denotes heuristic and (grey/dashed) random. Confidence intervals are given by $75\%$ and $25\%$ percentiles of the results from 30 random graphs, generated as explained in the main text.

to the limited accuracy of numerical integration. Additional synthetic experiments investigating the effect of different concentration parameters $c$ can be found in the supplementary C.1

**Incomplete data.** Next, we test our method for network inference from incomplete data. Noisy incomplete observations were generated by measuring the state at $N_s = 10$ uniformly random time-points and adding Gaussian noise with zero mean and variance $0.2$. Because of the expectation-step in Algorithm 1, is only approximate [11], we do not expect a perfect classifier in this experiment. We compare the exhaustive search, with a $K = 4$ parents greedy search, such that both methods have the same search-space. We initialized both methods with $\pi_i^{(0)}(m) = 1$ if $m = \mathrm{par}_{\mathcal{G}}(i)$ and 0 else, as a heuristic. In Figure 2 a) and b), it can be seen that both methods approach AUROCs and AUPRs close to one, for increasing amounts of data. However, due to the additional approximation in the greedy algorithm, it performs slightly worse. In Figure 2 c) and d) we plot the corresponding ROC and PR curves for 40 trajectories.

**Scalablity.** We compare the scalability of our gradient-based greedy structure search with a greedy hill-climbing implementation of structure seach ($K = 2$) with variational inference as in [11] (we limited this search to one sweep over families). We fixed all parameters as before and the number of trajectories to 40. Results are displayed in Figure 3.

**Dependence on initial values.** We investigate the performance of our method with respect to different initial values. For this, we draw the initial values of mixture components uniformly at random, and then project them on the probability simplex via normalization, $\tilde{\pi}_i^{(0)} \sim U(0,1)$ and $\pi_i^{(0)}(m) = \tilde{\pi}_i^{(0)}(m)/\sum_n \tilde{\pi}_i^{(0)}(n)$. We fixed all parameters as before and the number of trajectories to 40. In Figure 2, we displayed ROC e) and PR f) for our heuristic initial and random initial values. We find, that the heuristic performs almost consistently better.

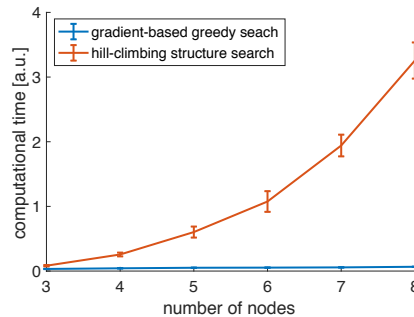

Figure 3: Run-time comparison of hill-climbing structure search with variational inference as in [11] with our gradient-based method.

Table 1: AUROC (AUPR) of different methods on IRMA-data (top performers in bold).

| method | | switch on | switch off |
|---|---|---|---|
| steady state | knockout | 0.68 (0.42) | 0.81 (0.50) |
| DBN | G1DBN | 0.78 (0.64) | 0.61 (0.34) |
| | VBSSM | 0.79 (0.70) | 0.76 (0.60) |
| ODE | TNSI | 0.68 (0.51) | 0.68 (0.42) |
| NDS | GP4GRN | 0.73 (0.61) | 0.76 (0.57) |
| | $CSI^d$ | 0.63 (0.46) | 0.86 (0.72) |
| | $CSI^c$ | 0.64 (0.39) | 0.73 (0.59) |
| GC | GCCA | 0.71 (0.55) | 0.74 (0.65) |
| CTBN | exhaustive | 0.81 (**0.86**) | **0.93** (**0.92**) |
| | greedy K=2 | **0.88** (0.85) | 0.91 (0.89) |
| random | | 0.65 (0.45) | 0.65 (0.45) |

## 5.2 Real-world data

**British household dataset.** We show scalability in a realistic setting, we applied our method to the British Household Panel Survey (ESRC Research Centre on Micro-social Change, 2003). This dataset has been collected yearly from 1991 to 2002, thus consisting of 11 time-points. Each of the 1535 participants was questioned about several facts of their life. We picked 15 of those, that we deemed interpretable, some of them, "health status", "job status" and "health issues", having non-binary state-spaces. Because the participants had the option of not answering a question and changes in their lives are unlikely to happen during the questionnaire, this dataset is strongly incomplete. Out of the 1535 trajectories, we picked

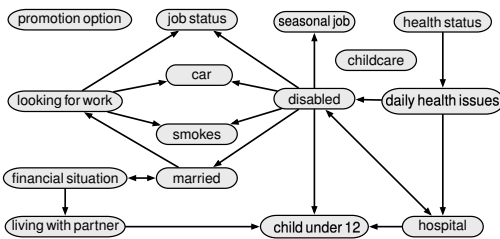

Figure 4: Learned structure using gradient-based greedy structure learning with maximal $K = 2$ parents from 600 trajectories.

600 at random and inferred the network presented in Figure 4. In supplementary C.2 we investigate the stability of this result. We performed inference with our greedy algorithm ($K = 2$). This dataset has been considered in [16], where a network among 4 variables was inferred. Inferring a large network at once is important, as latent variables can create spurious edges in the network [2].

**IRMA gene-regulatory network.** Finally, we investigate performance on realistic data. For this, we apply it to the *In vivo Reverse-engineering and Modeling Assessment* (IRMA) network [3]. It is, to best of our knowledge, the only molecular biological network with a ground-truth. This gene regulatory network has been implemented on cultures of yeast, as a benchmark for network reconstruction algorithms. Special care has been taken to isolate this network from crosstalk with other cellular components. The authors of [3] provide time course data from two perturbation experiments, referred to as "switch on" and "switch off", and attempted reconstruction using different methods. In Table 1, we compare to other methods tested in [18]. For more details on this experiment and details on other methods, we refer to the supplementary C.3, respectively.

## 6 Conclusion

We presented a novel scalable gradient-based approach for structure learning for CTBNs from complete and incomplete data, and demonstrated its usefulness on synthetic and real-world data. In the future we plan to apply our algorithm to new bio-molecular datasets. Further, we believe that the mixture likelihood may also be applicable to tasks different from structure learning.

## Acknowledgements

We thank the anonymous reviewers for helpful comments on the previous version of this manuscript. Dominik Linzner and Michael Schmidt are funded by the European Union's Horizon 2020 research and innovation programme (iPC–Pediatric Cure, No. 826121). Heinz Koeppl acknowledges support by the European Research Council (ERC) within the CONSYN project, No. 773196, and by the Hessian research priority programme LOEWE within the project CompuGene.

## Footnotes

[1]https://git.rwth-aachen.de/bcs/ssl-ctbn

[2]An over-complete graph has more edges than the underlying true graph, which generated the data.

[3]e.g. `https://string-db.org/` or `https://www.ebi.ac.uk/intact/`

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
