[Supplementary Material · gradient_ctbn_neurips_supplementary.pdf]

## Supplement: Scalable Structure Learning of Continuous-Time Bayesian Networks from Incomplete Data

In this supplementary, we give detailed descriptions on algorithms, data processing and derivations. Further, we provide an additional comparison to other methods for network reconstruction. All equation references point to the main text.

## A   Likelihood of Mixture of CIMs: Complete Data

### A.1   Likelihood of mixture of CIMs

We start with the log-likelihood of a CTBN

$$\ln P(\mathcal{M}, \mathcal{T} \mid R) = \sum_{i=1}^{N} \sum_{x} \sum_{u} \sum_{y \neq x} \{ M_i(x, y, u) \ln (R_i(x, y, u)) - T_i(x, u) R_i(x, y, u) \}$$

We choose to represent the rates as an expectation over different parent sets

$$R_i(x, y, u) = \sum_{m \in \mathcal{P}(\mathrm{par}_\mathcal{G}(i))} \pi_i(m) R_i(x, y, u_m)$$

$$\ln P(\mathcal{M}, \mathcal{T} \mid R) = \sum_{i=1}^{N} \sum_{x} \sum_{u} \sum_{y \neq x} \left\{ M_i(x, y, u) \ln \left( \sum_{m \in \mathcal{P}(\mathrm{par}_\mathcal{G}(i))} \pi_i(m) R_i(x, y, u_m) \right) \right.$$

$$\left. - T_i(x, u) \left( \sum_{m \in \mathcal{P}(\mathrm{par}_\mathcal{G}(i))} \pi_i(m) R_i(x, y, u_m) \right) \right\}$$

We apply Jensens inequality

$$\ln \left( \sum_{m \in \mathcal{P}(\mathrm{par}_\mathcal{G}(i))} \pi_i(m) R_i(x, y, u_m) \right) \geq \sum_{m \in \mathcal{P}(\mathrm{par}_\mathcal{G}(i))} \pi_i(m) \ln (R_i(x, y, u_m)),$$

and define (as in the main text) $M_i(x, y, u_m) \equiv \sum_{u/u_m} M_i(x, y, u)$ and $T_i(x, u_m) \equiv \sum_{u/u_m} T_i(x, u)$. This finally yields

$$\ln P(\mathcal{M}, \mathcal{T} \mid R) \geq \sum_{i=1}^{N} \sum_{m \in \mathcal{P}(\mathrm{par}_\mathcal{G}(i))} \pi_i(m) \sum_{x} \sum_{u_m} \sum_{y \neq x} \{ M_i(x, y, u_m) \ln (R_i(x, y, u_m)) - T_i(x, u_m) R_i(x, y, u_m) \}$$

$$= \sum_{i=1}^{N} \mathsf{E}_i^\pi \left[ \sum_{x} \sum_{u_m} \sum_{y \neq x} \{ M_i(x, y, u_m) \ln (R_i(x, y, u_m)) - T_i(x, u_m) R_i(x, y, u_m) \} \right].$$

We note that Jensen's inequality becomes sharp if $\pi_i(m) = 0 \; \forall m \neq m*$.

### A.2   Marginal Likelihood of Mixture of CIMs

Assuming independent priors $R_i(x, y, u_m) \sim \mathrm{Gam}(\alpha(x, y, u_m), \beta(x, y, u_m))$ we can calculate a marginal likelihood via

$$P(\mathcal{M}, \mathcal{T} \mid \pi) = \int dR \quad P(\mathcal{M}, \mathcal{T} \mid \pi, R) P(R \mid \alpha, \beta).$$

With the lower bound to the mixture likelihood,

$$P(\mathcal{M}, \mathcal{T} \mid \pi, R) \geq \prod_{i=1}^{N} \prod_{m} \prod_{u_m} \prod_{x,y} \exp \left[ \pi_i(m) \{ M_i(x, y, u_m) \ln (R_i(x, y, u_m)) - R_i(x, y, u_m) T_i(x, u_m) \} \right]$$

$$= \prod_{i=1}^{N} \prod_{m} \prod_{u_m} \prod_{x,y} R_i(x, y, u_m)^{\pi_i(m) M_i(x,y,u_m)} \exp \left[ -\pi_i(m) R_i(x, y, u_m) T_i(x, u_m) \right]$$

we can calculate a lower bound to the marginal mixture likelihood

$$
\begin{aligned}
P(\mathcal{M}, \mathcal{T} \mid \pi) \geq \int dR \quad & \prod_{i=1}^{N} \prod_{m} \prod_{u_m} \prod_{x,y} R_i(x, y, u_m)^{\pi_i(m)M_i(x,y,u_m)+\alpha(x,y,u_m)-1} \\
& \times \exp\left\{-\left(\pi_i(m)T_i(x, u_m) + \beta(x, y, u_m)\right)R_i(x, y, u_m)\right\}.
\end{aligned}
$$

Using the identity $\mathcal{I} = \int_0^{\infty} q^{M+a-1}e^{-(T+\tau)q}\mathrm{d}q = (T+\tau)^{-(M+a)}\Gamma(M+a)$, we can solve this integral analytically

$$
\begin{aligned}
P(\mathcal{M}, \mathcal{T} \mid \pi) \geq \prod_{i=1}^{N} \prod_{m} \prod_{u_m} \prod_{x,y} & (\pi_i(m)T_i(x, u_m) + \beta(x, y, u_m))^{-\{\pi_i(m)M_i(x,y,u_m)+\alpha(x,y,u_m)\}} \\
& \times \Gamma\left(\pi_i(m)M_i(x, y, u_m) + \alpha(x, y, u_m)\right).
\end{aligned}
$$

# B   Likelihood of Mixture of CIMs: Incomplete Data

## B.1   (Marginal) likelihood: Incomplete Data

In [2] an approximation to the likelihood of a CTBN given incomplete data was derived via an expansion in a coupling parameter $\varepsilon$

$$
\begin{aligned}
\ln P(\mathcal{D} \mid R, \mathcal{G}) \geq \sum_{i=1}^{N} \underbrace{\sum_{x}\sum_{u}\sum_{y \neq x}\left\{\mathsf{E}_q[M_i(x,y,u)]\ln\left(R_i(x,y,u)\right) - \mathsf{E}_q[T_i(x,u)]R_i(x,y,u)\right\}}_{\equiv E_i} \\
+ \sum_{i=1}^{N} H_i + \mathsf{E}_q[\ln P(\mathcal{D} \mid X)] + o(\varepsilon),
\end{aligned}
$$

with the parameter independent entropy

$$
H_i = \int_0^T \mathrm{d}t \sum_{x,u}\sum_{y \neq x} \tau_i(x, y, u; t)\left[1 - \ln\frac{\tau_i(x, y, u; t)}{q_i(x; t)q_i^u(t)}\right],
$$

and marginals and expected statistics as defined in the main text. We notice the similarity between the parameter-dependent part $E_i$ and the exact likelihood of a CTBN given complete data. By the exact same calculation as in the case of complete data, one arrives at

$$
\ln P(\pi \mid \mathcal{D}, \alpha, \beta) \geq \sum_{i=1}^{N} \mathcal{F}_i[\mathcal{D}, \pi, q] + \sum_{i=1}^{N} H_i + \mathsf{E}_q[\ln P(\mathcal{D} \mid X)] + \ln Z + o(\varepsilon)
$$

$$
\begin{aligned}
\mathcal{F}_i[\mathcal{D}, \pi, q] \equiv \sum_{m, u_m, x, x' \neq x} & \left\{\ln\Gamma\left(\bar{\alpha}_i^q(x, x' \mid u_m)\right) - \bar{\alpha}_i^q(x, x' \mid u_m)\ln\bar{\beta}_i^q(x \mid u_m)\right\} \\
& + \ln \mathrm{Dir}(\pi_i \mid c_i),
\end{aligned}
$$

with the updated posterior parameters $\bar{\alpha}_i^q(x, x' \mid u_m) \equiv \pi_i(m)\mathsf{E}_q[M_i(x, x' \mid u_m)] + \alpha_i(x, x' \mid u_m)$ and $\bar{\beta}_i^q(x \mid u_m) \equiv \pi_i(m)\mathsf{E}_q[T_i(x \mid u_m)] + \beta(x \mid u_m)$.

## B.2   Computing expected sufficient statistics via Euler-Lagrange Equations

Using Stirlings approximation $\Gamma(z) = \sqrt{\frac{2\pi}{z}}\left(\frac{z}{e}\right)^z + \mathcal{O}\left(\frac{1}{z}\right)$ we can approximate

$$
\mathcal{F}_i[\mathcal{D}, \pi, q] = \sum_{m}\sum_{u_m}\sum_{x, x' \neq x} \bar{\alpha}_i^q(x, x' \mid u_m)\left\{\ln\frac{\bar{\alpha}_i^q(x, x' \mid u_m)}{\bar{\beta}_i^q(x \mid u_m)} - 1\right\} + \ln \mathrm{Dir}(\pi_i \mid c_i)
$$

We can now derive Euler–Lagrange equations that maximize

$$\mathcal{L}[\mathcal{D}, \pi, q, \lambda] = \sum_{i=1}^{N} \left[ \mathcal{F}_i[\mathcal{D}, \pi, q] - \sum_{x, x' \neq x, u} \int_0^T dt\, \lambda_i(x; t) \left\{ \frac{d}{dt} q_i(x; t) - [\tau_i(x,' x, u; t) - \tau_i(x, x', u; t)] \right\} \right].$$

We have to calculate the derivatives

$$\partial_{q_i(x;t)} E_i = \sum_m \sum_{u_m} \sum_{x' \neq x} q(u_m; t) \pi_i(m) \frac{\bar{\alpha}_i^q(x, x' \mid u_m)}{\bar{\beta}_i^q(x \mid u_m)}$$

$$\equiv \sum_m \sum_{u_m} \sum_{x' \neq x} q(u_m; t) R_i^\pi(x, x' \mid u) = \mathsf{E}_i^u[R_i^\pi(x, x' \mid u)]$$

$$\partial_{q_i(x;t)} E_j = \sum_{m: i \in m} \sum_{u_m} \sum_{x', y \neq x'} q_j(x'; t) q_j(u_m/i; t) \pi_i(m) \frac{\bar{\alpha}_i^q(x', y \mid u_{m/i})}{\bar{\beta}_i^q(x' \mid u_m)}$$

$$\equiv \sum_{x', y \neq x'} q_j(x'; t) \mathsf{E}_j^u[R_j^\pi(x', y \mid u) \mid i, x]$$

$$\partial_{q_i(x;t)} H_i = -\sum_u \sum_{x' \neq x} \frac{\tau_i(x, x', u; t)}{q_i(x; t)}, \quad \partial_{q_i(x;t)} H_j = -\sum_{x, u/i} \sum_{x', y \neq x', u\text{ß}} \frac{\tau_j(x', y, u; t)}{q_i(x; t)},$$

$$\partial_{\frac{dq_i(x;t)}{dt}} \mathcal{L}[\mathcal{D}, \pi, q] = -\lambda_i(x; t)$$

and jump conditions follow from $\partial_{q_i(x;t)} \mathsf{E}_q[\ln P(\mathcal{D} \mid X)] = \sum_k \delta(t, t_k) \ln P(\mathcal{D}_i(t_k) \mid x)$, for a factorized observation model. Thus, we arrive at the first Euler-Lagrange equation

$$\text{I}: \quad \frac{d\lambda_i(x; t)}{dt} = \sum_u \sum_{x' \neq x} \frac{\tau_i(x, x', u; t)}{q_i(x; t)} - \mathsf{E}_i^u[R_i^\pi(x, x' \mid u)] + \Psi_i(x; t) + \sum_k \delta(t, t_k) \ln P(\mathcal{D}_i(t_k) \mid x),$$

with

$$\text{I.A} \quad \Psi_i(y; t) = \sum_{j=1}^{N} \left\{ \sum_{x, x' \neq x} \frac{\tau_j(x, x', u; t)}{q_i(y; t)} - \sum_{x'} q_j(x'; t) \mathsf{E}_j^u[R_j^\pi(x, x' \mid u) \mid i, y] \right\}$$

For the derivatives, with respect to the variational transition matrix, we get

$$\partial_{\tau_i(x, x', u; t)} E_i = -\sum_m \pi_i(m) \ln \left( \bar{\beta}_i^q(x \mid u_m) \right) + \sum_m \pi_i(m) \ln \left( \bar{\alpha}_i^q(x, x' \mid u_m) \right)$$

and

$$\partial_{\tau_i(x, x', u; t)} H_i = \ln[q_i(x; t) q_i^u(t)] - \ln \tau_i(x, x', u; t).$$

This forms the algebraic equation

$$0 = \lambda_i(x'; t) - \lambda_i(x; t) + \ln[q_i(x; t) q_i^u(t)] - \ln \tau_i(x, x', u; t) - \sum_m \pi_i(m) \ln \left( \bar{\beta}_i^q(x \mid u_m) \right)$$

$$+ \sum_m \pi_i(m) \ln \left( \bar{\alpha}_i^q(x, x' \mid u_m) \right).$$

Thus by defining $\rho_i(x; t) \equiv \exp(-\lambda_i(x; t))$, we get

$$\tau_i(x, x', u; t) = q_i(x; t) q_i^u(t) \frac{\rho_i(x'; t)}{\rho_i(x; t)} \prod_m \left( \frac{\bar{\alpha}_i^q(x, x' \mid u_m)}{\bar{\beta}_i^q(x \mid u_m)} \right)^{\pi_i(m)}$$

And we define

$$\text{II}: \quad \tau_i(x, x', u; t) \equiv q_i(x; t) q_i^u(t) \tilde{R}_i^\pi(x, x' \mid u) \frac{\rho_i(x'; t)}{\rho_i(x; t)}.$$

Figure 1: Investigation of the effect of the Dirichlet prior for different concentrations $c$ on the normalized lower bound of the marginal posterior (5) as described in the main text. We plotted the sample mean (dashed) to indicate the trend.

Derivation with respect to the Lagrange multipliers recovers the master equation

$$\text{III}: \quad \frac{\mathrm{d}}{\mathrm{d}t} q_i(x;t) = \sum_{x' \neq x, u} \left[ \tau_i(x,'x,u;t) - \tau_i(x,x',u;t) \right].$$

Inserting the identity II for $\tau_i(x,y,u;t)$ into the other equations I, I.A and III, yields the Euler-Lagrange equations from the main text. In particular we get

$$\Psi_i(y;t) = \sum_{j=1}^{N} \sum_{x,x' \neq x} q_j(x;t) \left\{ \mathsf{E}_j^u \left[ \tilde{R}_j^\pi(x,x',M,T) \mid y \right] \frac{\rho_j(x';t)}{\rho_j(x;t)} - \mathsf{E}_j^u \left[ R_j^\pi(x,x' \mid u) \mid i, y \right] \right\}.$$

## C  Experiments

### C.1  Effect of Dirichlet prior

We investigated the effects of different Dirichlet prior parameters $c$ on the approximate marginal likelihood (5) in the main text. We ran an experiment on a minimal CTBN example (2 nodes with a bidirectional coupling) in Figure 1 . We plotted the normalized lower bound of the marginal posterior $\mathcal{F}_i$ (eq. 5) for the node $i = 1$ (color coded) vs the mixture probability $\pi_1[1]$ (x-axis) for the node of having one parent (left, $\pi_1[1] = 0$) and having no parent (right, $\pi_1[1] = 1$) and the dashed sample mean to indicate the trend. The y-axis denotes the number of trajectories. While for a large amount of trajectories the mass is allocated at the ground-truth $\pi_1^*[1] = 0$ for all concentration parameters $c$ in the Dirichlet prior, for a small number of trajectories $c$ can either force selection ($c = 2$) or force a mixture through the convexity of the profile ($c = 0$).

### C.2  British household data

As this dataset does not have groundtruth, we can only check whether our predicted network is stable. For this consistency check, we predicted networks for varying numbers of samples and show convergence in the Hamming distance in Figure 2 towards the network from the main text. Note that the Hamming distance converges at a non-zero value - indicating that there is some variability remaining due to local optima and the restricted search space of $K = 2$ parents.

### C.3  IRMA data

**Processing IRMA data.** In this section we present our approach of processing IRMA data. The IRMA dataset consists of expression data of genes, measured in concentrations, which are continuous. We can not capture continuous data using CTBNs, but need to map this data to a set of latent states. We identify two states *over-expressed* ($X = 1$) and *under-expressed* ($X = 0$) with respect to the *basal* (equilibrium) concentration $c_B$. This motivates the following observation model given the basal

Figure 2: Convergence of predicted networks from different numbers of subsamples von BHPS dataset.

concentration

$$P(Y \mid X = 1, c_B) = \begin{cases} 1/|Y_0| & , Y \geq c_B \text{ and } Y \leq Y_0 \\ 0 & , \text{else} \end{cases},$$

$$P(Y \mid X = 0, c_B) = \begin{cases} 1/|Y| & , Y < c_B \text{ and } Y \geq -Y_0 \\ 0 & , \text{else} \end{cases},$$

where we have to choose some $Y_0$, so that the likelihood is normalized. We set $Y_0$ to some large value $Y_0 \geq \text{argmax}_{|Y| \in \text{DATA}}$ as our method remains invariant under each choice.

We model the basal concentration itself is a random variable, which we assume is gaussian distributed. We can estimate the parameters of the gaussian distribution $\mu_B$ and $\sigma_B$ from the data. The marginal observation model is then acquired by integration

$$P(Y \mid X) = \begin{cases} 1 - \text{erf}((Y - \mu_B)/\sigma_B) & , X = 1 \\ \text{erf}((Y - \mu_B)/\sigma_B) & , X = 0 \end{cases}.$$

Given this observation model we can assign each measurement a likelihood and can process the data using our method. We note that other models for IRMA data can be thought of that may return better (or worse) results using our method.

**Comparsion to other methods for network reconstruction.** We compare our method to the methods for network reconstruction from time-series expression data considered in [3], see table in the main text. These tests have, in contrast to [1], been performed on the full IRMA network. We adopt the shorthands of this paper to refer to different methods. The methods are based on dynamic Bayesian networks (DBNs), ODEs (TNSI), non-parametrics (NDS) and Granger Causality (GC). For more details on these methods we refer to [3]. For our evaluation, we used the original data and evaluation script from the DREAM challenge (`http://wiki.c2b2.columbia.edu/dream/index.php/The_DREAM_Project.`).