[Reviews · NeurIPS 2019]

Reviewer 1



I like the approach. This paper describes something new for continuous-time structure estimation. While mixtures have been explored for structure estimation in other domains, they have not been applied here and there are non-trivial hurdles that were overcome to do so. This method seems to be scalable. However, this is not tested in the paper. It would be good to see some experiments that would demonstrate this, particularly as the BHPS data was down-sampled to 200 trajectories (from 1535) and it isn't clear why. The variational method used seems very similar to that of [4]. This paper should make the connection clearer. Finally, structural EM (SEM) (see Section 19.4.3 of Koller & Friedman or Friedman's original 1997 paper) from BNs has been applied to CTBNs before (see [16], for example, and it seems to be implemented in CTBN-RLE). While exact inference is not scalable, this could be used with the the variation inference of [4]. This would make a natural comparison, as it is a scalable existing alternative that also employs variational inference. Minor notes: - I think Equation 2 needs a "+1" inside the Gamma function, as well. - The last equation on page 2 of the supplementary material does not seem to account for the sqrt(2pi/z) part of Stirling's approximation (which has an apostrophe, please note).

Reviewer 2



Originality; the work sheds new light on CTBN models and how to learn them from data in the case when big data has to be managed. Furthermore, the formulation of the structural learning algorithm for complete and incomplete data is a relevant step to improve effectiveness of CTBNs. Quality; the submission is formal and sound. However, I have the following concerns: Pag. 2; formula (1), I would ask to explain why the likelihood misses the part related to permanence in any given state, Pag. 3; I read "The integral ca ..." which one ? I suggest it is better to clarify this point, even if I know it. Pag. 5; the assumption about the alpha>>1 is quite strong and I would kindly ask to better motivate, investigate and analyze its' impact on solutions. I think this could strongly limit the application of the proposed approach in case where you have few observations w.r.t the number of variables. I found some minor typos.. I also would like to know something about inference on the CTBN once you have learnt it from data, i.e. how are you computing filtering, smoothing, ...? Clarity; the paper is in general quite clear, even if some more details and examples to introduce the idea of mixtures could have helped the reader to better understand the proposed approach. Significance; the contribution, in my humble opinion, is relevant with specific reference to the research area of CTBNs. Furthermore, it can help improve results achieved in relevant application domains as finance, medicine and biology.

Reviewer 3



Summary: Within the manuscript, the authors extend the continuous time Bayesian Networks by incorporating a mixture prior over the conditional intensity matrices, thereby allowing for a larger class compared to a gamma prior usually employed over these. My main concerns are with clarity / quality as the manuscript is quite densely written with quite some material has either been omitted or shifted to the appendix. For a non-expert in continuous time bayesian networks, it is quite hard to read. Additionally, there are quite a few minor mistakes (see below) that make understanding of the manuscript harder. As it stands, Originality: The authors combine variational inference method from Linzner et al [11], with the new prior over the dependency structure (mixture). By replacing sufficient statistics with expected (according to the variational distribution) sufficient statistics the authors derive a gradient based scheme according to the approximation to the (marginal likelihood). Quality/Clarity: As said, my main concern is about clarity and to some degree therefore also quality. My main confusion arises from section 4 (partly also 3), as the overall scheme is opaque to me. This is mainly due to the fact that part of the derivation is shifted to the appendix. As a result, it is unclear to me, how the expected moments can be computed from \rho_i, q_i. It is said, that this can be done from 7, but there I need \tau_i, how do I get this, this is not explained. Also, the final solution to (9) in (10,11) does not depend on the observations Y anymore, how is this possible? Some minor things contributing to my confusion: - Line 75: "a posteriori estimate": This is not a posterior over structures, but a maximum marginal likelihood estimate. - Eq (5), line 114: I was wondering about the 'normalization constant'. First, I think, it should be mentioned, that it is constant wrt to \pi. Second, Z is not necessarily the normalization constant of the true posterior but the approximation to the normalization constant that one would obtain, if the lower bound of line 105 would be used as likelihood, correct? - Algorithm 1: is only mentioned two pages later and the references to equations don't make sense. Also this algorithm is not explained at all. - Line 127: ref [5] is actually EP not VI - Line 149: the shorthand is used later not there. - Line 161: psi (x,t): I guess this should depend on Y. As stated the overall inference scheme does not depend on the observations Y, that does not make sense. - line 168: why should constraint ensure that incorporate noisy observations. The whole section is opaque to me. - Figure 1: subfigure labeling is wrong - Experiment british household: the authors report ROC scores, but do not mention the classification problem they are trying to solve, what was the ground truth? Also, it seems odd to me, that childcare is not linked to children. Significance: The proposed method does improve the scaling of inferring the dependency structure (reported from 4 nodes to 11). However, other approaches as in were discarded as not being sufficiently accurate or being too data hungry. The quality of the likelihood approximation for example could be evaluated on a small toy-example and compared against sampling based approaches, or [11].

[Author Response · NeurIPS 2019]

We thank all reviewers for carefully reading our manuscript and supplement. We also want to thank all referees for pointing out some minor issues, which we will not address individually in this rebuttal, but which will be implemented in the next version of the manuscript.

**To reviewer 1** We thank the reviewer for the positive assessment. The purpose of the BHPS experiment is to demonstrate better scalability of our approach compared to previous work as, e.g., in [4], [11] or [16]. With the previous methods, it would not be possible to infer a network of 15 subnodes (as we did here). However, inferring CTBNs from incomplete data inherently requires substantial computational efforts, as each trajectory has to be processed individually. This is the reason why we down-sampled the BHPS data (as was also done in [16], but for far fewer subnodes). We will add a more detailed experiment in the next version of the manuscript. Here, we will vary the number of trajectories to investigate the stability of our prediction. For a discussion about the similarity of our approach with [4] we refer to [11], where the weak-coupling approximation (for non-mixture CTBNs) was derived. The reviewer is correct that a more scalable structure learning method than in [16] can be constructed in combination with [4] (or [11]). Structure estimation in CTBNs from incomplete data has two distinct bottlenecks: 1.) Latent state estimation (scales exponential in the number of nodes). 2.) Iteration over candidate structures (scales super-exponential in the number of nodes). While 1.) has been overcome in [4] and [11] to different orders of accuracy, here we are the first to touch on bottleneck 2.) by introducing mixtures of generators. Our method is inherently more scalable than SEM ([4] or [11]) due to the additional approximation step (Jensen's inequality in line 103-106). We show improved scalability in figure c), where we compare the run-time of the greedy version of our method with greedy hill-climbing (maximum number of parents is two) for SEM+variational inference [11]. We did account for the $\sqrt{2\pi/z}$ term. However, an approximation, which holds for $\bar{\alpha} \gg 1$, is omitted in the current version.

**To reviewer 2** We thank the reviewer for the positive assessment. For a discussion about the likelihood function, we would like to refer to [15] where the likelihood for a CTBN is derived. We would like to point out here that $\bar{\alpha} \gg 1$ is a quite natural assumption, even for situations where data is sparse. Here, a Bayesian perspective implies that sufficiently strong prior information has to be leveraged to regularize the problem and allow for inference. We exactly do this when assuming $\bar{\alpha} = M + \alpha \gg 1$ by incorporating a sufficiently strong prior $\alpha$ on the number of recorded transitions $M$. While inference in a learned CTBN is not our main focus in this manuscript, we employ smoothing under a mixture CTBN by solving eq. (10) and (11), to calculate the expected sufficient statistics. This algorithm can also be applied to the learned model, to perform inference. Based on the suggestions of the reviewer we ran an experiment on a minimal CTBN example (2 nodes with a bidirectional coupling) in figure $a$) given below. We plotted the normalized lower bound of the marginal posterior $\mathcal{F}_i$ (eq. 5) for the node $i = 1$ (color-coded) vs the mixture probability $\pi_1[1]$ (x-axis) for the node of having one parent (left, $\pi_1[1] = 0$) and having no parent (right, $\pi_1[1] = 1$) and the dashed sample mean to indicate the trend. The y-axis denotes the number of trajectories. While for a large amount of trajectories the mass is allocated at the ground-truth $\pi_1^*[1] = 0$ for all concentration parameters $c$ in the Dirichlet prior, for a small number of trajectories $c$ can either force selection ($c = 2$) or force a mixture through the convexity of the profile ($c = 0$).

**To reviewer 3** We thank the reviewer for the detailed review of our work. Based on his suggestions, we will improve the readability in the next version of our manuscript. The expression for $\tau_i$ can be found in the main text in line 164. The derivation was shifted to the supplement due to length constraints. To avoid confusion, we will feature this expression as an equation in the next version. The solutions of the ODEs (10,11) do depend on the data Y, as it is explained in lines 165-168 in the main text. Here, the observations are explained to be implemented via jump conditions on the Lagrange multipliers $\rho_i$ (line 168). This is in line with existing works, e.g., [4], [11] and [17] (in [17] it is derived in the main text). As mentioned in the main text, we solve the set of ODEs in the same way as in these works. The equations to be referenced in Alg. 1 are indeed (10), (11) and the jump condition on $\rho_i$ (line 168), thank you for pointing this out. We refer to [5] in line 127 as a variational approach. The authors of [5] themselves refer to their method as a variational approach, e.g., in their abstract. We did not report ROC scores for the BHPS, as no ground-truth is available (because the true dependencies are unknown). We ran this experiment only to demonstrate scalability. In case the reviewer is referring to Fig. 2 a) and b), these subfigures refer to a separate experiment on synthetic data in a larger system using greedy search, as explained in the main text. Based on the suggestion of the reviewer, we investigated the accuracy of the lower bound of the marginal posterior (see eq. 5) against the true marginal posterior (calculated via numerical integration) for mixtures of different entropies, different amounts of trajectories for $c = 1$, see figure $b$) below. This experiment is performed on the graph ensemble as in the first synthetic experiment of our main paper with rates drawn from a Gamma distribution ($\alpha = 5$ and $\beta = 10$). For scalability, we refer to our response to reviewer 1.



[Meta-Review · NeurIPS 2019]

This paper contributes a new technique for the estimation of structure in continuous time Bayesian networks, and completes the picture with an accompanying inference method and an illustration on a real-world problem. There is agreement among reviewers that this is a high quality contribution, if one takes the confidence-weighted scores from reviewers into account. As a point for improvement for the paper, we could reiterate a comment that was raised in the reviewer discussion: "[the paper] is missing reasonable and helpful experimental comparisons that are not hard to do, given that the code exists already in CTBN-RLE" and the authors are encouraged to consider broadening their experimental comparisons for a final published version.